# Fuzzy Neural Network PID Strategy Based on PSO Optimization for pH Control of Water and Fertilizer Integration

**Runmeng Zhou, Lixin Zhang \*, Changxin Fu, Huan Wang, Zihao Meng, Chanchan Du, Yongchao Shan and Haoran Bu**

School of Mechanical and Electrical Engineering, Shihezi University, Shihezi 832003, China; zrm18899129681@163.com (R.Z.); fcx623623@163.com (C.F.); wangh_cn@126.com (H.W.); mzh9823@163.com (Z.M.); dcc_1314@163.com (C.D.); shanyongchao@stu.edu.cn (Y.S.); buhaoran@stu.edu.cn (H.B.)
**\*** Correspondence: zhlx2001329@163.com

**Abstract:** In the process of crop cultivation, the application of a fertilizer solution with appropriate pH value is more conducive to the absorption of nutrients by crops. If the pH of the irrigation water and fertilizer solution is too high, it will not only be detrimental to the absorption of nutrients by the crop, but will also damage the structure of the soil. Therefore, the precise regulation of pH in water and fertilizer solutions is very important for agricultural production and saving water and fertilizer. Firstly, the article investigates the hybrid control of fertilizer and water conditioning systems, then builds a fuzzy preprocessing controller and a neural network proportional–integral–differential controller, and optimizes the neural network parameters by means of an improved particle swarm algorithm. The effectiveness of the controller was verified by comparison with the common proportional–integral–differential control and fuzzy algorithm control for fertilizer control and fuzzy preprocessing neural network control. Simulation experiments for this study were designed through the MATLAB/Simulink simulation platform, and the simulation results show that the algorithm has good tracking and regulation capabilities in the system. Finally, the four control algorithms are experimentally validated under different pH regulations using designed field experiments. The results show that, compared with other control algorithms, the control algorithm in this paper has a smaller overshoot and good stability with a shorter rise time, which can achieve the purpose of better regulating the fertilizer application system.

**Keywords:** water and fertilizer mixing; fuzzy processing; pH control; neural network PID control; particle optimization swarm algorithm



## 1. Introduction

Water and fertilizer integration technology is widely used in agricultural production, in which the concentration ratio and proper application of water and fertilizer are crucial to the growth of crops [1]. Currently, fertilizer and irrigation water use has gradually increased worldwide without a proportional increase in crop yields [2]. A qualitative analysis of fertilizer application methods shows that the main cause of this situation is the improper distribution of fertilizers, which affects the efficiency of fertilizer uptake and the normal growth of the crop at all stages. Excessive fertilizer application can also damage soil structure, affecting the growing environment and survival of crops [3–6]. Therefore, in order to enhance the rationality of crop growth and the utilization rate of water and fertilizer resources, and to ensure the normal development and application of agricultural resources, more accurate and reasonable water and fertilizer blending integration technology should be adopted to realize the transformation of agricultural technology from being crude to intensive.

The root systems of different crops require different levels of acidity and alkalinity, and most mono-textured fertilizers are alkaline; thus, the pH of the fertilizer needs to be

explicitly adjusted at the fertilizer blending stage [7,8]. However, the process of blending liquid fertilizer for pH is often affected by factors such as time lag and nonuniformity of fertilizer delivery, and the variation in liquid fertilizer pH is also nonlinear in nature. Therefore, fast and accurate adjustment of liquid fertilizer pH to set requirements is an important research area in precision agriculture [9,10]. The research on pH values mainly includes algorithm optimization research and fertilizer system pH model building research. The current modeling research on pH process regulation mainly includes linear and nonlinear models and artificial intelligence models [11]. In the study of optimization of nonlinear control, methods such as adaptive regulation, feedback linearized transformation control and sliding mode control are generally used. Considering that the traditional PID control strategy has difficulty dealing with the complex nonlinearity and time lag in the control of water and fertilizer environments, more intelligent algorithms for optimal control need to be investigated [12].

At this stage, most water and fertilizer controllers use conventional PID control algorithms internally. However, in the actual fertilizer application process, the fertilizer concentration and flow rate have a complex impact on the water–fertilizer mixing system, which leads to problems such as time-varying and lagging system control objects always existing. The use of PID control algorithms is usually not effective for regulation [13,14]. In view of this, the authors in [15] used the application of a fuzzy adaptive control to regulate the water–fertilizer system. The simulation results show that the fuzzy PID adaptive control method is able to track and regulate the dynamic changes of the nutrient preparation process more accurately than the traditional PID method. In [16], a single-neuron PID learning algorithm based on quadratic optimization was proposed for the strongly nonlinear pH process. The effectiveness of the algorithm was verified by the simulation control results of the strongly nonlinear pH process. In [17], a self-tuning fuzzy PID control algorithm was introduced into the control system to regulate the frequency of the fertilizer pump, and hence, the water and fertilizer through the fuzzy PID control algorithm. An expert system PID control was developed by the authors in [18]. The parameters of the controller can be adjusted according to the deviation of the pH value and the deviation rate. Simulation experiments showed that the expert PID control has good control performance. The authors in [19] studied and established a grey prediction model for water and fertilizer irrigation and used MATLAB software to simulate fuzzy PID control in irrigation and fertilizer application, which effectively improved the accuracy of water and fertilizer concentration according to experimental verification. The authors in [20] used quantitative feedback theory, a particle swarm optimization algorithm and a genetic algorithm to optimize PID control. The nonlinear problem of pH control was optimized. Additionally, the control performance was verified by simulation experiments. A wavelet-BP neural network-based method for accurate fertilizer application to maize was proposed in [21]. The accuracy of the optimal fertilizer amount was improved by establishing a wavelet-BP neural network to calculate the nonlinear problem of fertilizer application. The authors in [22] used an incremental PID algorithm to control the water and fertilizer ratios and constructed a simulation model with online parameter settings using RBF neural networks. According to the experimental verification, the RBF-PID model was more accurate and stable than the incremental PID model. The authors in [23] proposed a nonlinear model predictive control method based on an elastic BP neural network and hybrid grey wolf optimizer. Simulation results showed that the proposed controller performed well and could reduce the disturbance caused by nonlinearity.

In summary, the main work carried out in this paper to improve the accuracy and rapidity of fertilizer pH regulation, based on water and fertilizer integration, is as follows: (a) An improved neural network controller is proposed which improves control by fuzzifying the inputs and optimizing the network parameters. (b) Particle swarm algorithms are applied for the optimization of two-layer weight parameters of neural networks. (c) PID control, fuzzy control, fuzzy neural network PID control and the control proposed in this paper are simulated and verified on the MATLAB/Simulink platform, and it is concluded

that the algorithm proposed in this study is better than other control algorithms. (d) In order to verify the practicality and reliability of the proposed algorithm, the experimental verification of the data acquisition, the transfer function, and the upper computer reading and decision controls are designed in this paper, and the results prove that the control algorithm proposed in this paper has better results and performance compared with other control methods.

The paper is structured as follows. The first portion of this paper analyses the pH regulation process and develops a mathematical model of water and fertilization through the equilibrium equation. In the second part, a fuzzy preprocessing controller and a neural network controller are constructed, and the neural net controller is optimized using a particle swarm algorithm. The second part firstly constructs the controller that fuzzily preprocesses the input signal, then constructs the neural network controller, and finally uses the particle swarm algorithm to optimize the neural network controller. The third part compares the control effects of the four control algorithms through software simulation and experimental verification, and finally obtains the verification results based on the water and fertilizer system in this study.

## 2. pH Control Equipment and Control Model

### 2.1. Equipment System Components

The structure of the water and fertilizer integration control equipment is shown in Figure 1. When mixing fertilizer, the main control module with a computer as the core controls the opening and closing time of the solenoid valves at the outlets of the fertilizer tank, irrigation water source and acid tank, thus regulating the proportion of irrigation water and fertilizer in the mixing tank. A stirring pump is used to mix the water and fertilizer in the tank. The fertilizer tank is judged by a pH sensor to determine whether the pH value of the liquid inside meets the pH value for crop growth, and if not, the solenoid valve of the acid tank is opened to replenish the tank with acid. Fertilizer application is carried out mainly through the computer-based main control module, which controls the fertilizer pump and solenoid valve.

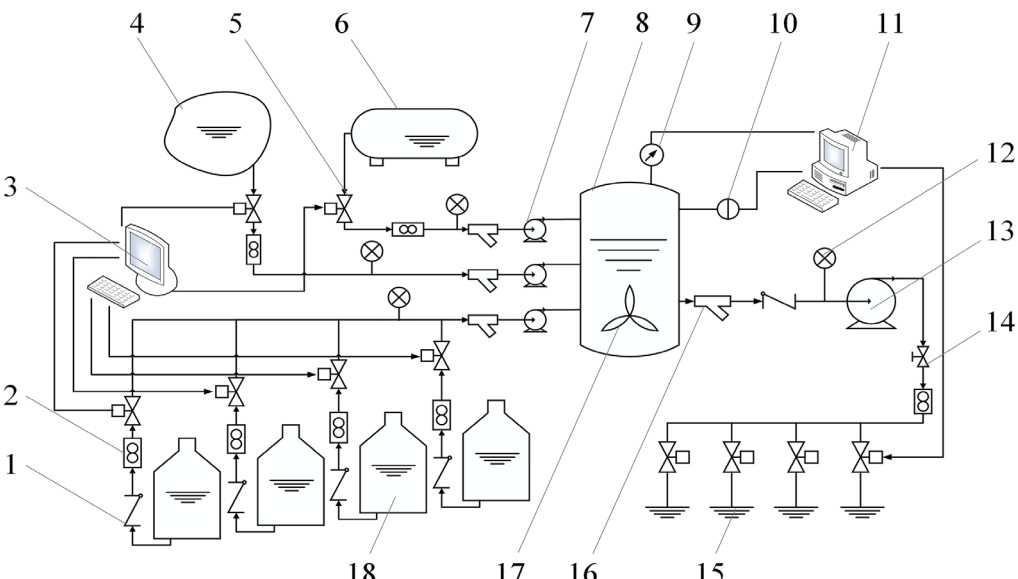

**Figure 1.** Water and fertilizer integration and control equipment structure diagram: 1—check valve; 2—flow meter; 3—computer control; 4—water source; 5—solenoid valve; 6—acid tanks; 7—hose pump; 8—mixing tank; 9—level meter; 10—pH sensor; 11—monitoring and adjustment of the upper side; 12—pressure gauge; 13—irrigation and fertilization pump; 14—holding valve; 15—drip irrigation belt; 16—filter; 17—stirring pump; 18—fertilizer tank.

### 2.2. Analysis of the Water and Fertilizer pH Adjustment Process

In order to obtain the water and fertilizer mixture, fertilizer and water are fed into the mixing tank on a cyclical basis. The configuration of aqueous fertilizer mixing solutions requires the periodic opening and closing of valves to feed fertilizer and water into the mixing tank. In the actual fertilizer mixing process, in addition to the process of agitation and mixing, delayed factors such as changes in the flow rate of the liquid in the pipeline and the use of the monitoring module can introduce time lags into the pH adjustment.

Both water and fertilizer are usually weakly alkaline; thus, an acidic regulating liquid is required to adjust the pH. When the three liquids of water, fertilizer and a regulating liquid, are mixed, it can be seen as a neutralization process between a strong acid and a weak base. The pH of the fertilizer in the mixture is made to be similar to the pH set by the system. The inherent nonlinearity of the acid–base neutralization process also has an impact on the pH adjustment process. As the pH adjustment in the actual fertilizer mixing process is influenced by a variety of variable disturbances, a simplified control model is developed to analyze it qualitatively from a mechanistic point of view. It is assumed that the fertilizer drum is fed with raw fertilizer, acid and pure water as inputs to the pH model so that the fertilizer is rapidly integrated with the water and that the concentration of the mixed solution is used as an output, and that the mixed fertilizer in the drum is always full and of the same concentration at the top and bottom, as shown in Figure 2.

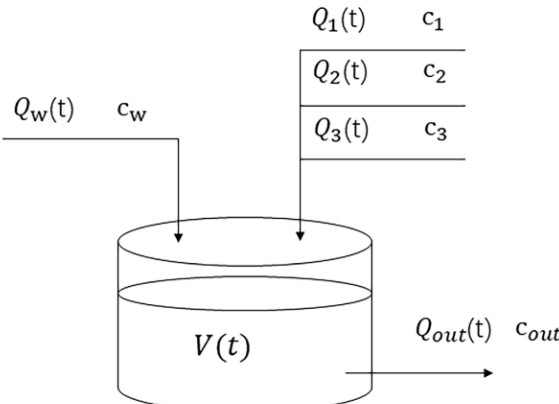

**Figure 2.** Fertilizer pH adjustment concentration analysis chart.

According to the material balance principle and the law of atomic conservation, the equilibrium state of the system should comply with the conservation of fertilizer and volume conservation, as to construct the soil conductivity and acidity concentration control process equation.

$$\left\{ \begin{array}{c} V\frac{dH_1}{dt} = L_a H_a - L_1 H_1 \\ V\frac{dx_2}{dt} = L_n H_n + L_w H_w - L_1 H_2 \\ L_1 = L_n + L_w + L_a \\ 10^{-pH} - 10^{pH-14} - H_1 + \frac{1}{1+10^{pK_b+pH-14}}H_2 = 0 \end{array} \right\} \tag{1}$$

where $V$ is the volume of mixed liquid, $L_a$ is the acid input flow, $C_a$ is the acid concentration, $L_n$ is the fertilizer input flow, $H_n$ is the fertilizer concentration, $L_w$ is the water input flow, $H_w$ is the water input concentration, $L_{out}$ is the output flow of the system, $H_1$ is the concentration of acid in the mixture, $H_2$ is the concentration of alkali in the mixture, $K_b$ is the weak-base ionization equilibrium constant, $pK_b = -\log(K_b)$, pH is the output variable of the process, $K_b$ is the ionization constant of the weak base and $K_w$ is the ionization constant of water ($10^{-14}$).

*2.3. System Model Building*

Considering the control characteristics of the liquid fertilizer pH value control system comprehensively, the first-order system transfer function and delay link are used to describe the mathematical model of the fertilizer solution's pH value. The parameters to be identified in the model include the system gain *K*, delay time $\tau$ and time constant *T*. The transfer function is shown in Equation (2):

$$G(s) = \frac{K}{Ts+1}e^{-\tau s} \tag{2}$$

Given the step response of pH = 6.5 as the input signal of the open-loop system, the sampling time interval is set to 1 s, the initial pH value of the mixed liquid is 7.6 and the step response curve of the system is fitted using the first-order approximation method of the computer-fitted system according to the data change of the pH value, then it is identified that the system gain *K* is 0.56, the delay time $\tau$ is 4 and the time constant *T* is 39. The pH regulation process has a time lag.

## 3. Construction of the Control Strategy

*3.1. PID Control Algorithms*

The block diagram of a conventional PID control system is shown in Figure 3. PID control is the differential between the actual output signal of the control object and the given signal for making differences and tracking. The control rate u of the system is derived from the proportional P, the integral I and the differential D as a means of regulating the performance indicators of the fertilizer. However, due to the effects of time lag and nonlinearity in the fertilizer mixing process, the effect of using a PID control strategy to regulate pH is often unsatisfactory and it is difficult to track the changes in the pH of the fertilizer solution [24].

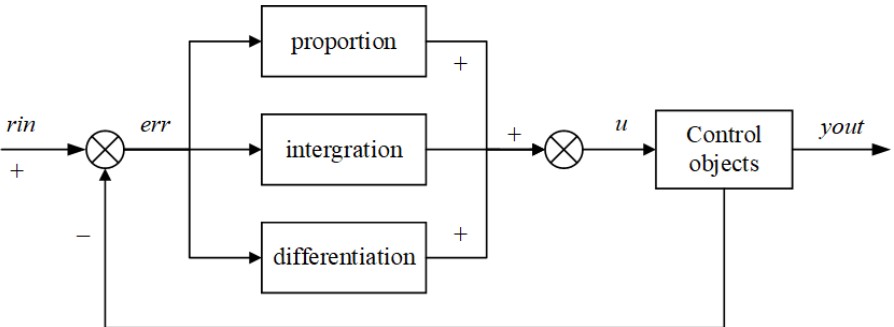

**Figure 3.** PID control schematic.

*3.2. Fuzzy-BPNN-PID Controller Design*

Compared with the PID control algorithm, the fuzzy control algorithm has the advantages of good robustness, fault tolerance, etc. The operator can set appropriate fuzzy rules to regulate the output based on the existing knowledge and experience of fertilizer regulation, and the input signal of the system is targeted to facilitate the control link to achieve the structured regulation requirements; however, the fuzzy algorithm relies on simpler fuzzy rules due to the lack of the self-learning capability, which leads to lower control accuracy and poor dynamic quality of the system, and adaptive regulation cannot be achieved [25].

Compared with the above two algorithms, the BP neural network control algorithm has the feature of adaptive learning, meaning that it can learn and adjust adaptively for the input, complete the adjustment of its own network parameters, and finally, realize the optimal adjustment of the output signal [26]. The block diagram of the BPNN-PID controller is shown in Figure 4.

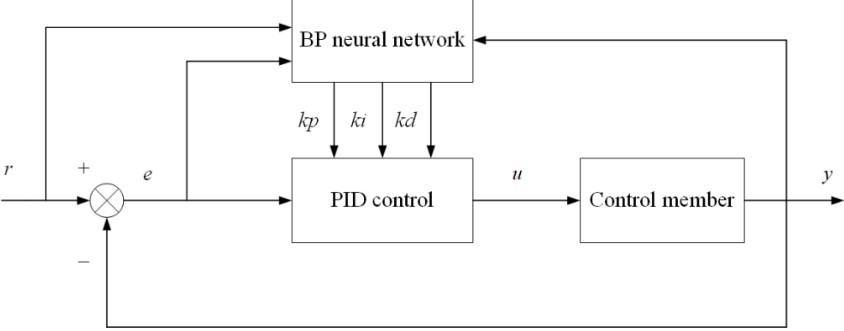

**Figure 4.** Schematic diagram of neural network PID control.

In order to obtain better control performance, combined with the advantages of the above fuzzy algorithm and BP neural network control algorithm, we chose to use fuzzy control to fuzzify and normalize the system state variables *e* and *ec* in advance, before using the BP neural network, to avoid the situation that the output is saturated due to the excessive direct input when the neural network activation sigmoid function is sampled, and reduce the situations where the network output is no longer sensitive to the input [27]. Then, the neural network was used to learn and control the output signal of the fuzzy module, as to achieve the purpose of accurately adjusting the pH value of water and fertilizer.

3.2.1. Design of Fuzzy Preprocessing Controller

In this paper, The error of pH E and the deviation of change of pH error EC were selected as input variables for the input of the fuzzy preprocessing control module, and the fuzzed system output $v(k)$ was taken as the output variable. The role of this module is to fuzzify and normalize the system state variables *e* and *ec*. The error of pH *e*, the deviation of change of pH error *ec* and the output quantity $v(k)$ of the fuzzification processing module all adopt the triangular membership function, and the area centroid method was chosen as the clearing method of the fuzzy controller. The controller sets the fuzzy linguistic values for both the pH deviation *e* and the rate of change of deviation *ec* of water and fertilizer as {NB, NM, NS, ZO, PS, PM, PB}, where NB, NM, NS, ZO, PS, PM and PB represent negative large, negative medium, negative small, 0, positive small, positive medium and positive large, respectively. The fuzzy domain takes on the value of the eigenpoints corresponding to {−6, −4, −2, 0, 2, 4, 6}. The fuzzy language values of the output signal $v(k)$ of the fuzzy controller are set as {NB, NS, ZO, PS, PB}, and the fuzzy theoretical domain takes the value points {−4, −2, 0, 2, 4}. The fuzzy control rules for the controller output are shown in Table 1. The fuzzified output $v(k)$ is determined according to the fuzzy theoretical domain and the affiliation function, and then the obtained values are sent to the input layer of the neural network. The subordination functions of the input and output of the fuzzy processing and the fuzzy surface diagrams between them are shown in Figures 5–7.

**Table 1.** Fuzzy rule tables based on fuzzy control.

| E \ EC | NB | NM | NS | O | NS | NM | NB |
|---|---|---|---|---|---|---|---|
| NB | NB | NB | NM | NM | NM | NB | NB |
| NM | NB | NB | NM | NM | NM | NB | NB |
| NS | NM | NM | NS | NS | NS | NM | NB |
| O | NM | NM | NS | NS | NS | NM | NM |
| PS | NM | NM | NS | NS | NS | NM | NM |
| PM | NB | NB | NM | NM | NM | NB | NB |
| PB | NB | NB | NM | NM | NM | NB | NB |

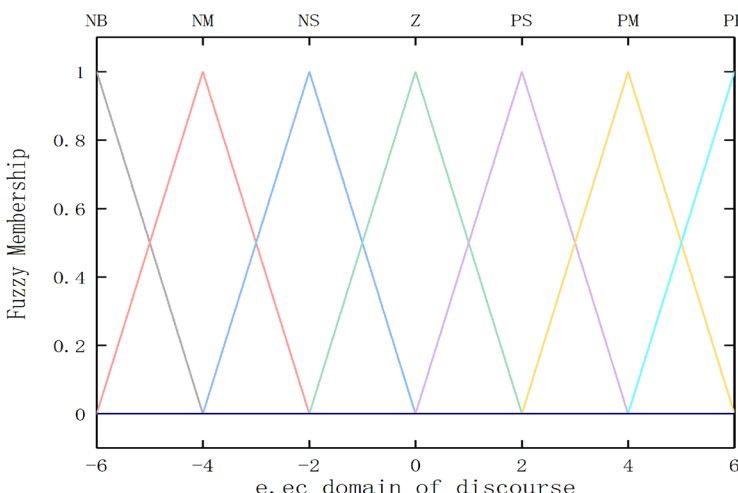

**Figure 5.** Plot of the error and the affiliation function for the rate of change of the error.

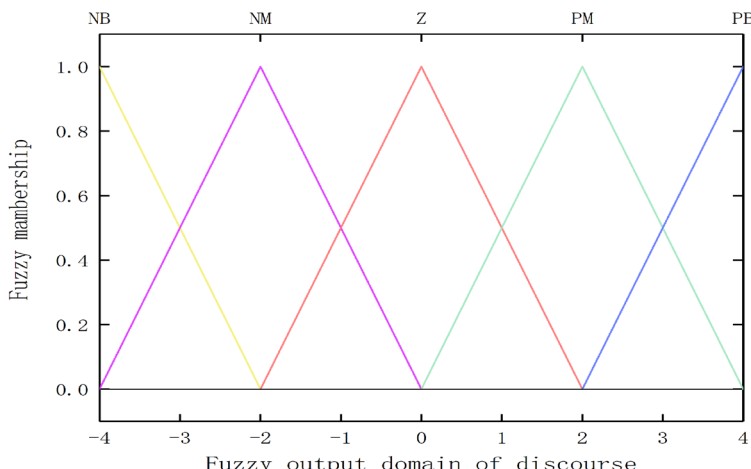

**Figure 6.** Fuzzy output $v(k)$ subordinate degree function.

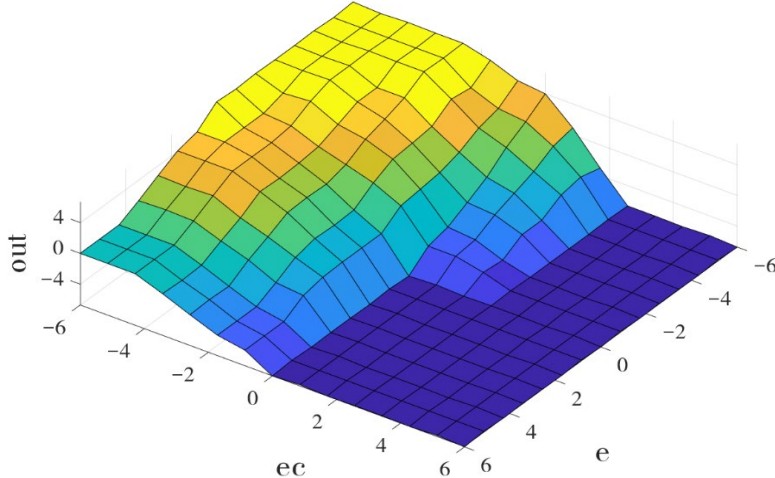

**Figure 7.** Fuzzy control surface diagram.

In Figure 5, each triangle is part of the affiliation function. For example, the blue triangles represent input variables e and ec with a theoretical domain between −4 and 0 and output linguistic variables of NS. The work is based on the theoretical domain of the input quantities and the area centre of gravity method to determine their output linguistic

variables. In Figure 6, each triangle is used to determine the output linguistic variable for the fuzzy variable $v(k)$. For example, the red triangle indicates that the theoretical domain of $v(k)$ is between $-2$ and $2$ and the output linguistic variable is Z. Figure 7 represents a surface plot of the fuzzy algorithm design, from which it can be seen that when e and ec are small, the output is in the yellow peak portion; when e and ec are large, the output is in the blue low portion; and when the values of e and ec are in the middle of the theoretical domain, the output is in the green stable portion.

### 3.2.2. Design of BP Neural Network PID Controller

A BP neural network is a multi-layer, feed-forward neural network, which contains three layers of a network structure, namely, the input layer, hidden layer and output layer. The number of nodes in the input and output layers is determined by the dimensionality of the input vector and output vector, respectively, and the hidden layer plays an important role in the function and structure of the network [28].

The neural network controller in this paper used a three-layer BP neural network, the inputs to which are fuzzy-processed system state variables. The network contains three input layer nodes, nine implicit layer nodes and three output layer nodes. The structure of the fuzzy preprocessed neural network is shown in Figure 8. The role of each layer of the neural network and the input–output relationship between the layers are as follows.

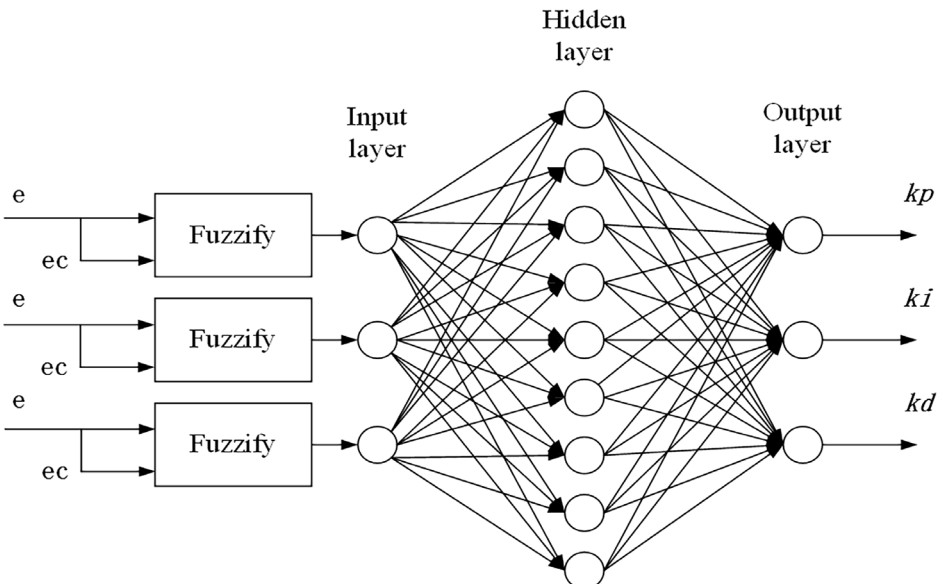

**Figure 8.** Fuzzy-BPNN-PID structure diagram.

The first layer is the input layer, where the input is the system state variable after fuzzy processing; the second layer is the hidden layer, and the input is:

$$\begin{cases} net_i^{(2)}(k) = \sum\limits_{j=1}^{3} \omega_{ij}^{(2)} O_j^{(1)}(k) \\ O_j^{(2)}(k) = f\left[net_i^{(2)}(k)\right] \\ (i = 1, 2 \cdots \cdots 9) \end{cases} \tag{3}$$

where $\omega_{ij}^{(2)}$ is the implied layer weighting factor. $f\left[\cdot\right]$ is the hidden layer activation function, selected as $f(x) = \tan h(x)$.

The final layer is the output layer, with the following inputs and outputs:

$$
\begin{cases}
net_{li}^{(3)}(k) = \sum_{i=1}^{9} \omega_{li}^{(3)} O_i^{(2)}(k) \\
O_l^{(3)}(k) = g\left[net_l^{(3)}(k)\right] \\
\quad (l = 1, 2, 3) \\
O_1^{(3)}(k) = k_p \\
O_2^{(3)}(k) = k_i \\
O_3^{(3)}(k) = k_d
\end{cases}
\tag{4}
$$

where $\omega_{li}^{(3)}$ is the output layer weighting factor. $f\left[\cdot\right]$ is the output layer activation function, selected as $g(x) = e^x / (e^x + e^{-x})$.

After determining the neural network outputs $k_p$, $k_i$ and $k_d$, they are brought into the incremental PID formula for calculation. The incremental PID control is able to remember and maintain the state of the system at the previous moment, thus making the impact of possible errors minimal; therefore, incremental PID is used to obtain the optimal control signal, and the incremental PID formula is:

$$
\begin{cases}
\Delta u(k) = K_P[e(k) - e(k-1)] + K_I e(k) \\
\quad + K_D[e(k) - 2e(k-1) + e(k-2)] \\
\quad\quad u(k) = u(k-1) + \Delta u
\end{cases}
\tag{5}
$$

where $\Delta u(k)$ is the control signal increment and $k$ is time; $k_P$ is the proportionality factor; $k_i$ is the integration factor; and $k_d$ is the differentiation factor.

## 4. Improved PSO-Optimized Fuzzy-BPNN Control Algorithm

The learning process of a BP neural network consists of continuously adjusting the weighting parameters of each layer of the network and then using the weighting coefficients to calculate the optimal control parameters. Therefore, the BP neural network needs to learn and continuously update the weighting coefficient matrix of each layer of the network.

### 4.1. BP Neural Network Weight Coefficient Update

According to the BP learning algorithm, first define the objective cost function $J$:

$$
J = \tfrac{1}{2}e^2(k) = \tfrac{1}{2}[r(k) - y(k)]^2
\tag{6}
$$

After determining the target cost function $J$, perform a negative gradient direction search with the objective of minimizing the target function $J$, with an additional inertia term that accelerates the convergence of the search to a global minimum as follows:

$$
\Delta\omega_{li}^{(3)}(k+1) = -\eta \frac{\partial u(k)}{\partial \omega_{li}^{(3)}} + \alpha \Delta\omega_{li}^{(3)}(k)
\tag{7}
$$

where $\eta$ is the learning rate; $\alpha$ is the inertia coefficient.

The correction formula for the weighting coefficient of the output layer of the BP neural network is further obtained as:

$$
\begin{cases}
\Delta\omega_{li}^{(3)}(k+1) = \eta\delta_l^{(3)} O_i^{(2)}(k) + \alpha\Delta\omega_{li}^{(3)}(k) \\
\delta_l^{(3)} = e(k+1)sgn\left(\frac{\partial y(k+1)}{\partial u}\right) \times \frac{\partial u(k)}{\partial O_l^{(3)}(k)} \times g'\left[net_l^{(3)}(k)\right] \\
\quad\quad (l = 1, 2, 3)
\end{cases}
\tag{8}
$$

Based on the above derivation, the correction formula for the implied layer weighting factor can be obtained as follows:

$$
\begin{cases}
\Delta\omega_{ij}^{(2)}(k+1) = \eta\delta_i^{(2)}O_j^{(1)}(k) + \alpha\Delta\omega_{ij}^{(2)}(k) \\
\delta_i^{(2)} = f'\left[net_i^{(2)}(k)\right]\sum_{l=1}^{3}\delta_l^{(3)}\omega_{li}^{(3)}(k)(i=1,2,\cdots 9)
\end{cases}
\tag{9}
$$

which sets up: $g'(x) = g(x)[1 - g(x)]$; $f'(x) = [1 - f^2(x)]/2$.

Therefore, the workflow of the Fuzzy-BPNN-PID controller can be summarized as follows: Set the initial values of the network parameters of each layer of the BPNN, as well as the learning rate and inertia coefficient of the network; use the fuzzy controller to fuzz the acquired $e$ and $ec$; input the fuzzified output $v(k)$ to the BPNN; then, calculate the input and output of each layer of neurons through the BPNN; and finally, calculate the three adjustable parameters $k_p$, $k_i$ and $k_d$ of the PID controller, which are calculated by Equation (5). The output control signal is $u(k)$. The system continuously updates $u(k)$ and the network weight parameters according to the updates of $e$ and $ec$. The control schematic is shown in Figure 9.

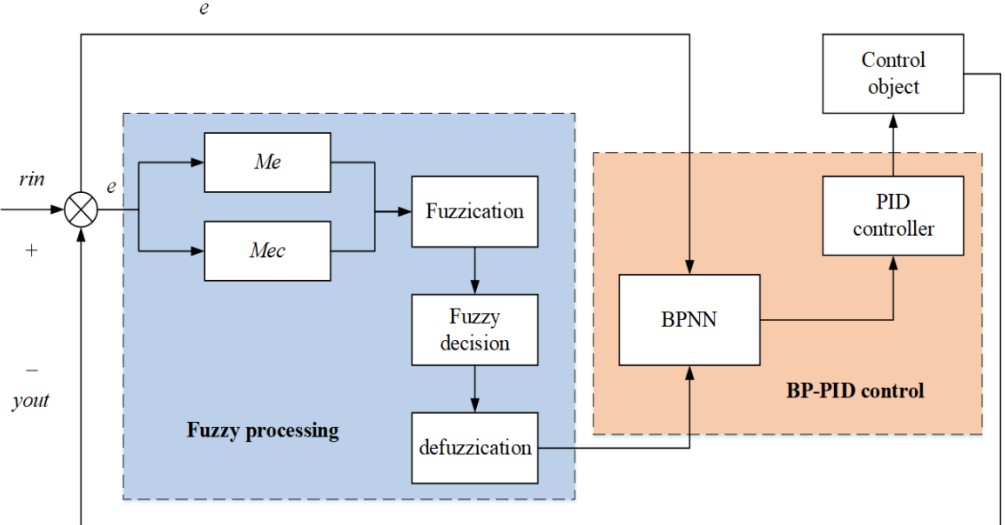

**Figure 9.** Fuzzy-BPNN-PID algorithm control schematic.

*4.2. Improved Particle Swarm Algorithm*

The optimization search process of the Fuzzy-BPNN-PID controller is as follows: firstly, the fuzzy rules are used to preprocess the two input signals $e$ and $ec$; then, the fuzzy processed and normalized signals are fed into the neural network for training and learning; and finally, the optimal output control signal and the corresponding network weight parameters are obtained. However, the initial weight parameters for each layer of the BP neural network are usually chosen as random numbers of $[-1, 1]$, and choosing random numbers as the initial values causes the algorithm to converge slowly, greatly affecting the training efficiency of the network and causing the algorithm to converge to local extremes [29]. Therefore, in order to obtain better network training accuracy, this paper used the particle swarm algorithm to calculate and find the optimal parameters as the initial network weights of the BP neural network, as to improve the iterative convergence speed of the network and the efficiency of finding the optimal control signal.

4.2.1. Standard Particle Swarm Algorithm

The particle swarm algorithm (PSO) is an algorithm for population intelligence optimization. It works by constructing a population of M particles in a D-dimensional space, where each particle represents a potential optimal solution to the desired value. The system

sets each particle ($i$ = 1, 2, . . . , M) to have a position parameter $x_i$ and a velocity parameter $v_i$, as well as a fitness value obtained according to the objective function. The particle evaluates the individual optimal position $P_{id}^k$ and the population optimal position $G_{id}^k$ of the particle based on the fitness value, and each time the particle is updated, it obtains a new fitness value as well as $P_{id}^k$ and $G_{id}^k$, thus gradually finding the historical optimal position $P_{id}^k$ of the particle and the historical optimal position $P_{gd}^k$ of the particle population [30]. The updated iterative equation is calculated as:

$$V_{id}^{k+1} = \omega V_{id}^k + c_1 r_1 \left( P_{id}^k - X_{id}^k \right) + c_2 r_2 \left( P_{gd}^k - X_{id}^k \right) \tag{10}$$

$$X_{id}^{k+1} = X_{id}^k + V_{id}^{k+1} \tag{11}$$

where $\omega$ is the inertia weight; $d$ = 1, 2, . . . , D; $i$ = 1, 2, . . . , M; $V_{id}$ is the velocity of the particle; $c_1$ and $c_2$ are acceleration factors; and $r_1$ and $r_2$ are random numbers in the interval [0, 1].

When using the PSO algorithm for parameter optimization, the performance evaluation index of the fitness function needs to be specified. The purpose of the algorithm in this paper is to reduce the value of deviation; therefore, this paper selected the time multiplied by the absolute value error integral criterion index as the fitness function $J$ of the particle swarm algorithm, and the calculation formula is:

$$J = \int_0^n t \cdot |e(t)| dt \tag{12}$$

where $n$ is the total number of iterative steps of the particle swarm.

### 4.2.2. Improved Particle Swarm Algorithm

The inertia weight $\omega$ has a balanced role in the global search for the optimal and development direction of the particle swarm algorithm and has an important impact on the performance of the algorithm. The adjustment of the inertia weight has a strong correlation with the number of iterations of the algorithm, and the inertia weight tends to decrement nonlinearly with the increase in the number of iterations; if the particles enter the optimal solution range during the iteration but the inertia coefficient does not reach the expected value due to the decrement, the particles deviate from the optimal. If the particle enters the optimal solution range during the iteration but the inertia coefficient does not reach the expected value due to decreasing, the particle deviates from the limit of the solution [31]. In order to make up for the shortcomings of the inertia weight parameter adjustment strategy, this paper improves the inertia weights of the particles with the expression:

$$\omega = \omega_{min} - \frac{(\omega_{max} - \omega_{min}) \times (f_i - f_z)}{3(f_a - f_z)} \tag{13}$$

where $\omega_{max}$ is the maximum value of the set inertia weight; $\omega_{min}$ is the minimum value of the set inertia weight d; $f_i$ is the current fitness value of the ith particle; $f_a$ is the average fitness value of the population particles; and $f_z$ is the optimal fitness value of the current particle.

Equation (13) shows that the value of $\omega$ is dynamically adjusted according to the difference between the particle runtime fitness value and the optimal fitness, thus improving the performance of the algorithm.

### 4.3. Fuzzy-BPNN Optimal Control Algorithm Based on Improved Particle Swarm

The main principle of the improved particle swarm-based Fuzzy-BPNN-PID control algorithm is that the initial weights of the BP neural network can be optimized accurately and quickly by introducing an improved particle swarm algorithm with an inertia weight adjustment strategy to improve the search efficiency of the network. At the same time, the state variables at the input of the neural network are fuzzily preprocessed to reduce the output of the network, which is insensitive to the input and easy to saturate. The steps are as follows:

(1) Initialization of particle swarm, where parameters such as population size, particle dimension, number of iteration steps, initial inertia weight and learning factor are set according to neural network factors.

(2) Using the global search capability of the particle swarm algorithm, the approximate optimal solution of the network weight parameters is preferentially obtained, and the particles are updated according to the particle fitness. If the particle swarm meets the requirements of the optimal fitness value, the corresponding network parameters are transmitted to the BP network.

(3) The BP neural network uses the network weights of the improved particle swarm optimization to process the fuzzy preprocessed signal, updates the network parameters through back propagation, and finally, outputs the optimal $k_p$, $k_i$ and $k_d$, which are calculated according to the incremental PID formula that also calculates the control signal $u(k)$. Its control principle diagram is shown in Figure 10.

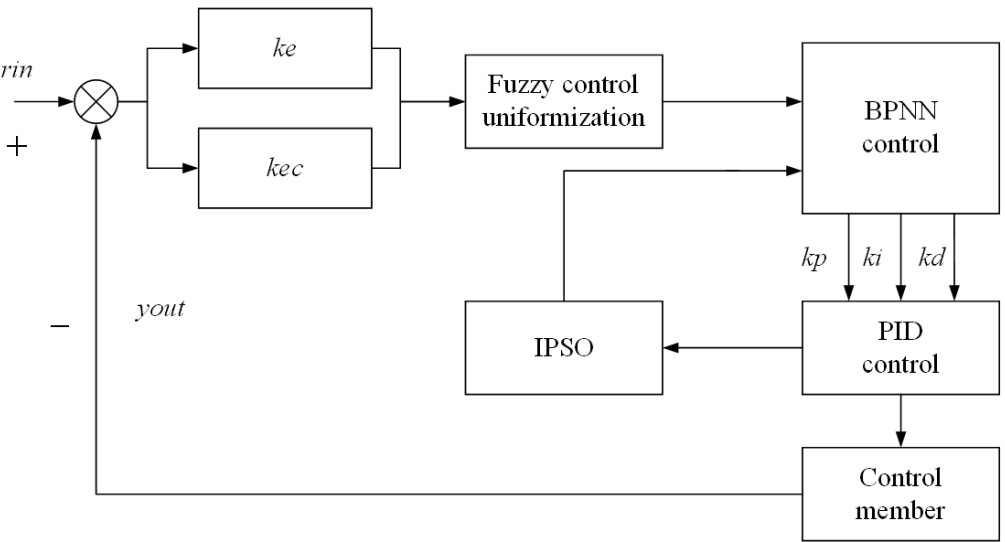

**Figure 10.** IPSO-Fuzzy-BPNN-PID system schematic diagram.

When the IPSO algorithm is used to obtain the network weight parameter matrix, the particle swarm population size is M = 20, the number of iteration steps is N = 100, the learning factor is $c_1 = c_2 = 2$, the initial inertia weight is $\omega_0 = 0.9$, the particle velocity interval is set to $[-5, 5]$, the structure of the neural network in this paper is 3-9-3 and the number of adjustable network parameters is $2 \times 3 \times 9 = 54$; therefore, the particle dimension D = 54 is set, and each dimension represents a variable-adjusted parameter.

## 5. Simulation Experiment of pH Control of Fertilizer Solution

In order to verify the performance of the composite control scheme, the Simulink simulation interface of the fuzzy controller was first designed through MATLAB, and then the BP-PID control algorithm based on PSO optimization was written using the S-function module of the Simulink module. PID control, fuzzy control, BPNN-PID control with fuzzy preprocessing and BPNN-PID control system models based on improved PSO and fuzzy preprocessing were established in MATLAB/Simulink, as shown in Figure 11. The workflow of the model running is shown in Figure 12. The pH value of pure water was 8, the acid used to adjust pH was dilute hydrochloric acid with a concentration of 0.2 mol/L, the flow rate of water entering the mixing tank was 1.1 L/s and the system delay time was 4 s.

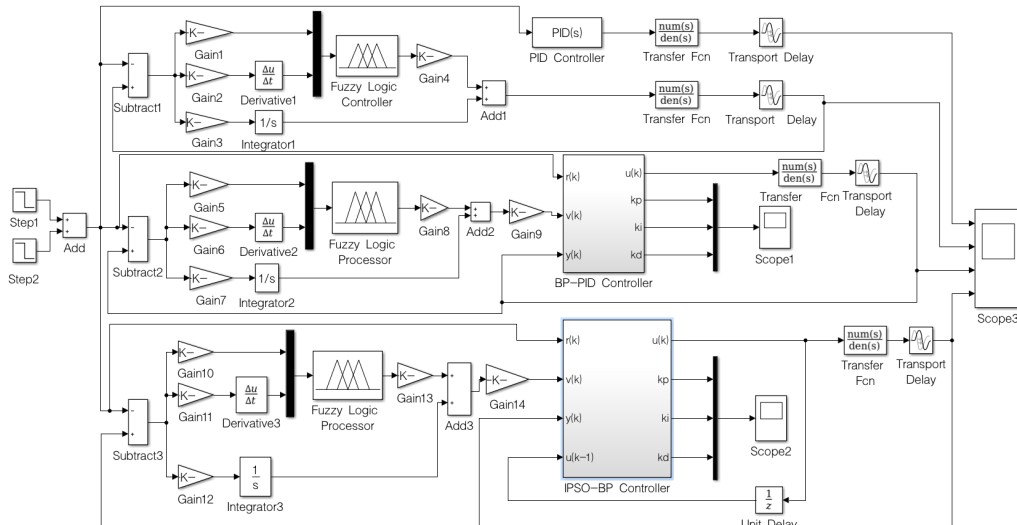

**Figure 11.** Simulation model of fertilizer–liquid mixing pH control system.

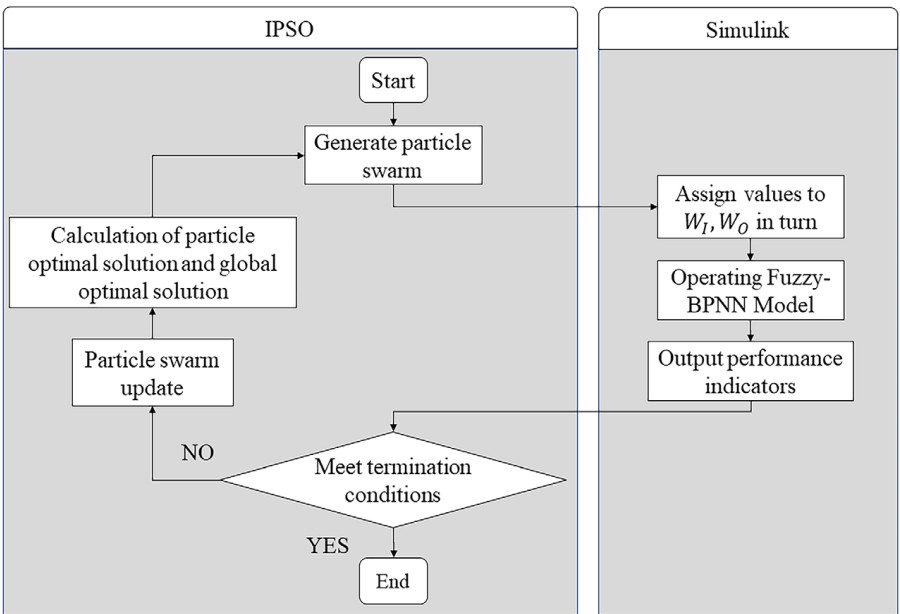

**Figure 12.** The flowchart of proposed controller.

The pH value of the fertilizer solution was tracked with a simulation time of 1000 s; the tracking curves of different control algorithms are shown in Figure 13. According to Figure 13, it was found that the PID controller has a fast response but a large overshoot, the fuzzy control has a small overshoot but a slow rise time, the fuzzy neural network controller has a faster response time but there is still a certain amount of overshoot, and the IPSO-Fuzzy-BPNN-PID controller has a faster response time with a small overshoot and can respond to changes in the set pH value in a short time. Compared with the other three control algorithms, it is more conducive to improving the accuracy of the fertilization process. In theory, this composite controller could optimize the pH regulation of fertilizers and water. In order to further analyze the optimization performance of the controller in this paper, the process of adjusting the pH value of the water and fertilizer mixture from 7.5 to 6.5 was simulated according to the actual situation. The simulation time was 500 s. The control curves of several control algorithms are shown in Figure 14. According to Figure 14, the fuzzy control and PID control have the largest response time and overshoot, respectively, and the fuzzy neural network response is faster, but there is a certain amount

of overshoot over a period of time. The performance parameters of the four algorithms are shown in Table 2. From Table 2, it was found that in terms of overshoot amount, it is reduced by 7.0, 2.0 and 1.7 percentage points compared to the first three algorithms. The steady times are reduced by 60 s, 235 s and 88 s. The IPSO-Fuzzy-BPNN-PID controller performed well in terms of overshoot and response time, and was able to reach the set pH value in a short time. In the next phase, the algorithm is tested.

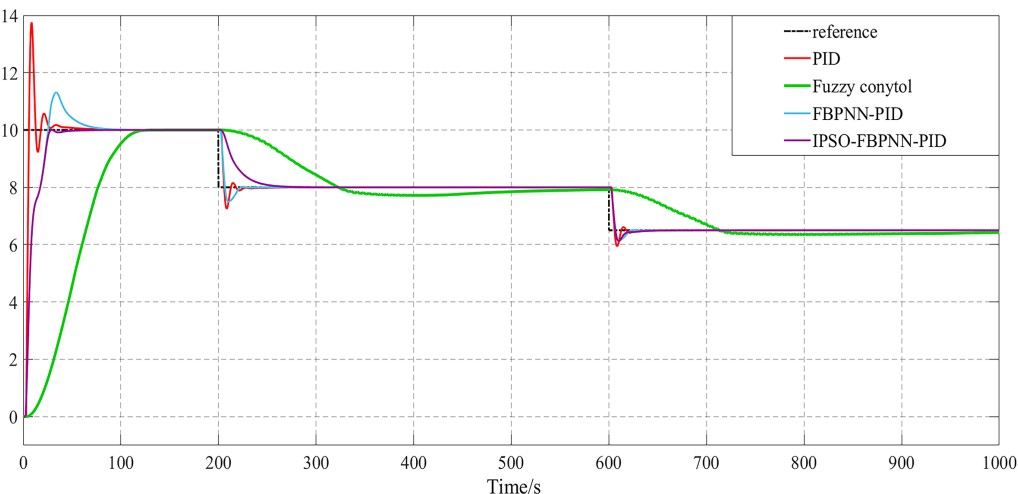

**Figure 13.** Simulation comparison and analysis diagram of different controllers.

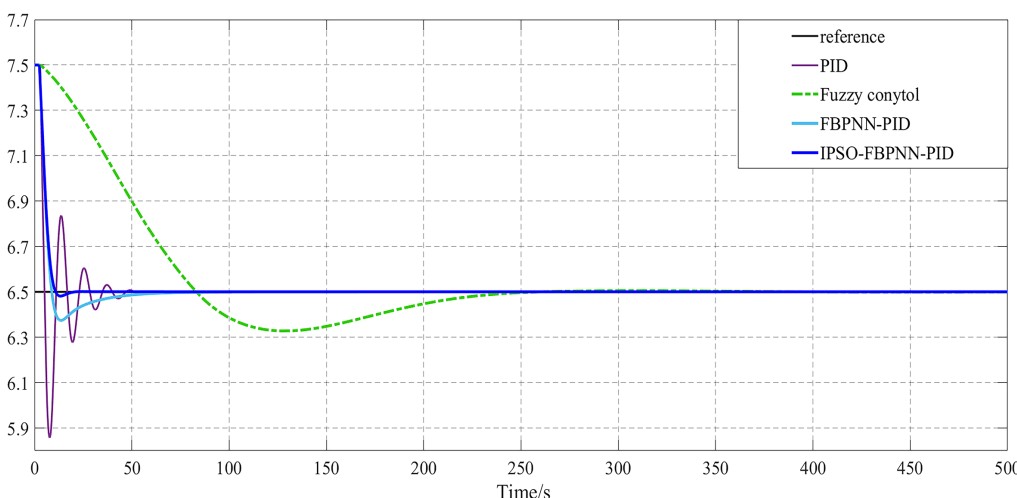

**Figure 14.** Control analysis and comparison chart of different controllers.

**Table 2.** Comparison table of performance parameters of four control algorithms.

| The Controller Type | Rise Time (s) | Peak Time (s) | Steady-State Time (s) | Maximum Overshot (%) |
|---|---|---|---|---|
| PID | 5 | 9 | 80 | 7.9% |
| Fuzzy | 80 | 122 | 255 | 2.9% |
| FBPNN-PID | 9 | 14 | 108 | 2.6% |
| IPSO-FBPNN-PID | 11 | 13 | 20 | 0.9% |

## 6. Tests and Analysis

In order to confirm the stability of the algorithm, intelligent fertilization trials were carried out. The schematic diagram of the test device structure is shown in Figures 15 and 16.

The test system structure mainly consisted of a main control module, a signal transmission module, a sensor monitoring module and a drive module. The main control module was mainly composed of a computer, PLC controller and touch screen configuration interface. The signals were transmitted via an RS485 bus. The drive control module was divided into the fertilizer-proportioning drive module and irrigation control drive module. The tests took the set pH value as input value and used PLC as the control core to receive the actual feedback signal from the pH sensor and perform calculations. When blending the fertilizer, the system master control module controlled the fertilizer output frequency of the fertilizer-proportioning drive module by changing the analogue voltage signal, controlled the input volume of the regulating liquid as it was added to the mixing tank according to the calculated volume and completed the mixing, and the system maintained a stable state when the pH reached the set value.

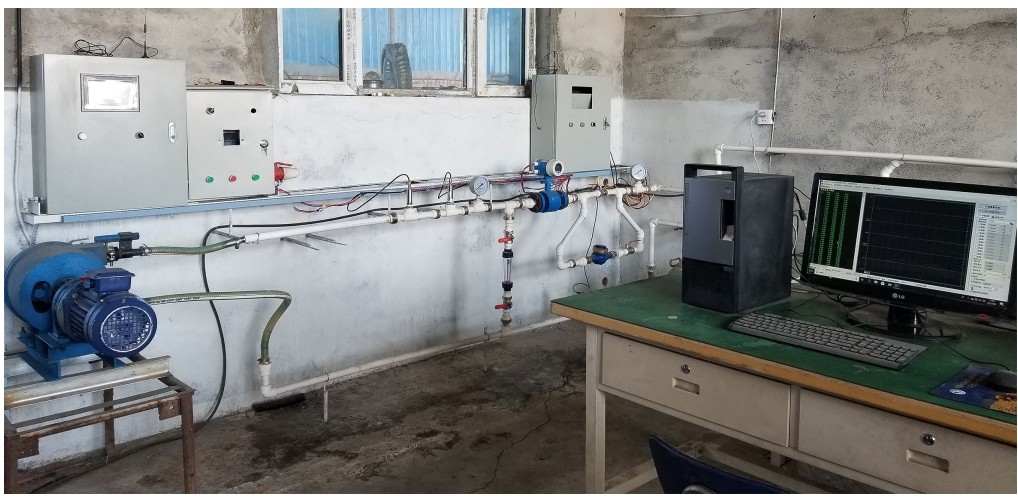

**Figure 15.** Fertilizer pH adjustment and detection experimental platform.

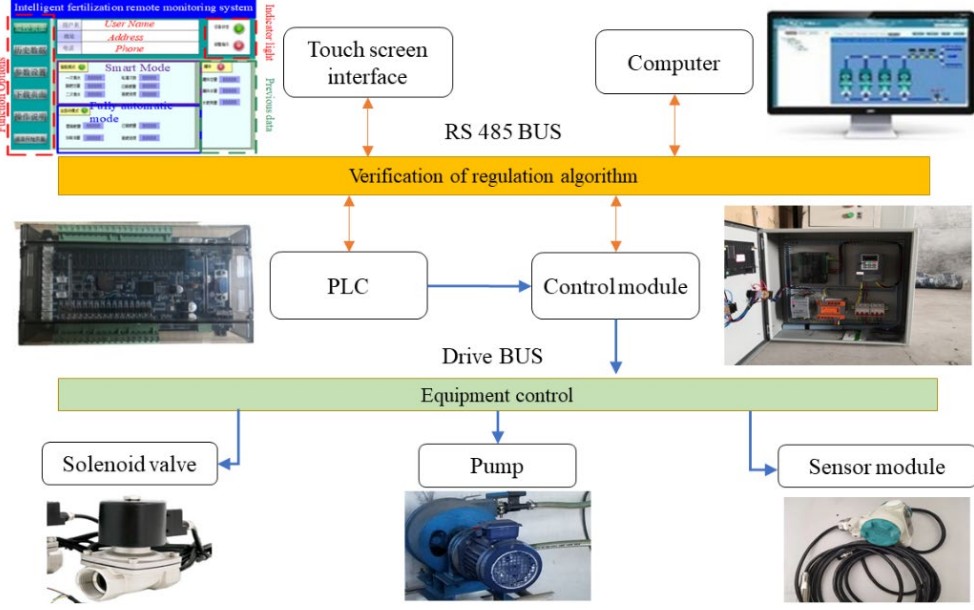

**Figure 16.** Structure of the fertiliser pH adjustment test system.

The control algorithm used float accuracy with a sampling period of 2 s. Two groups of pH adjustment processes were set up; the initial pH values of the two water and fertilizer regulation tests were 7.1 and 6.9, respectively, the target pH values of the water and fertilizer

regulations were adjusted to 6.3 and 5.3, respectively, and an electromagnetic flow meter was selected to determine the instantaneous flow rate. The variable control test was read by sensors and data acquisition cards to test and compare different control strategies, setting the system flow rate to 4 m³/h. The purpose of conducting this test was to regulate the accuracy of fertilizer pH control by the four controllers. The results of the test comparison are shown in Figures 17 and 18. The indicator performance of the four controllers is shown in Tables 3 and 4. As can be seen from Figures 17 and 18, as the fertilizer flow rate increased, the performance of all four controllers gradually improved. Although the PID controller rose faster, it had the highest overshoot and a longer oscillation time, with a larger difference to the set pH value in a short time. The fuzzy controller had a smaller overshoot, but the response time was slow and could not keep track of the set pH value in time. The fuzzy neural network controller was considerably higher in rise time, overshoot and adjustment time compared to the first two controllers, but it had a significantly higher overshoot than the controller proposed in this paper, with the performance parameters tabulated as shown in Table 3. When the pH value was reduced from 7.1 to 6.3, the control proposed in this paper was the smallest in both overshoot and steady-state time. In terms of overshoot, the reductions compared to the first three algorithms were 8.1, 4.6 and 1.5 percentage points. In terms of steady time, the reductions were 39 s, 120 s and 34 s.

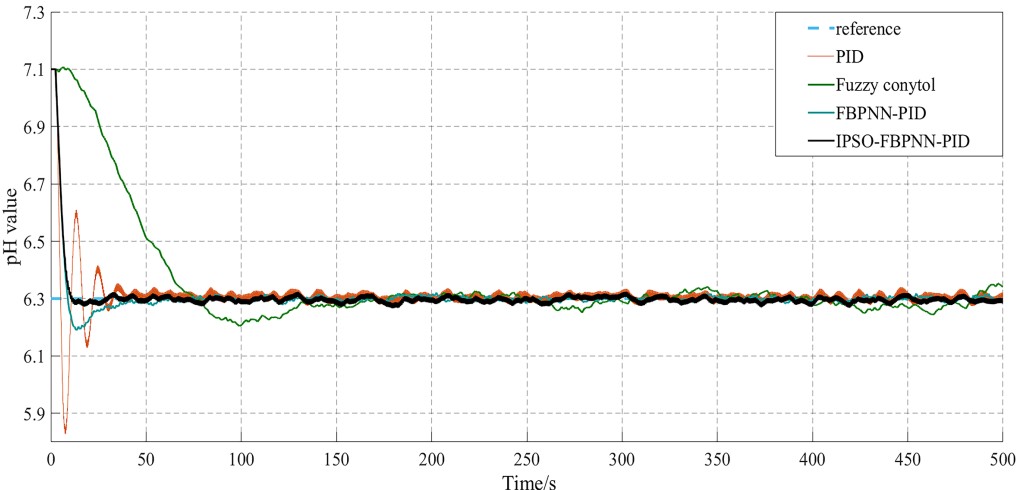

**Figure 17.** Comparative analysis of different control algorithms when regulating pH from 7.1 to 6.3.

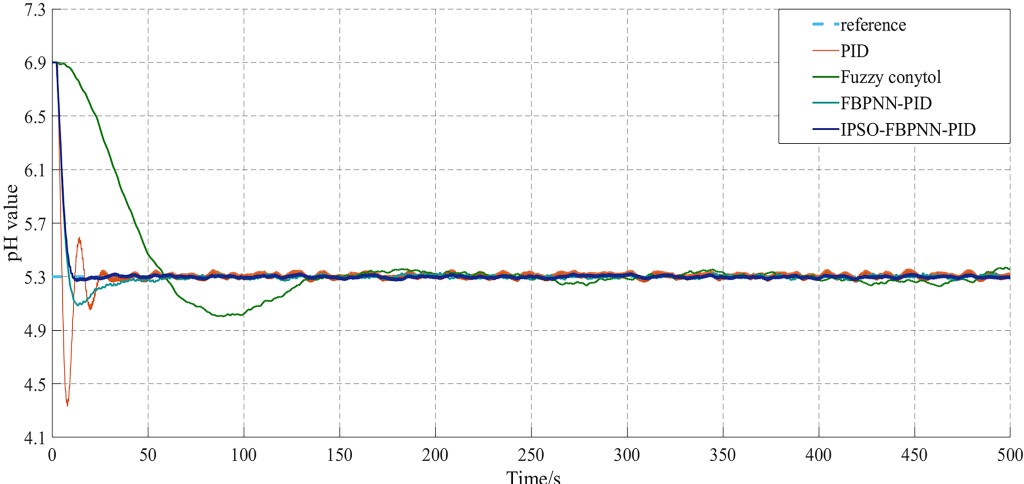

**Figure 18.** Comparative analysis of different control algorithms when regulating pH from 6.9 to 5.3.

**Table 3.** Table of performance parameters for pH from 7.1 to 6.3.

| The Controller Type | Rise Time (s) | Peak Time (s) | Steady-State Time (s) | Maximum Overshot (%) |
|---|---|---|---|---|
| PID | 5.5 | 7.7 | 52 | 8.2% |
| Fuzzy | 75 | 99 | 133 | 4.7% |
| FBPNN-PID | 8.4 | 15 | 47 | 1.6% |
| IPSO-FBPNN-PID | 10 | 12 | 13 | 0.1% |

**Table 4.** Table of performance parameters for pH from 6.9 to 5.3.

| The Controller Type | Rise Time (s) | Peak Time (s) | Steady-State Time (s) | Maximum Overshot (%) |
|---|---|---|---|---|
| PID | 6 | 8.4 | 42 | 13.8% |
| Fuzzy | 58 | 88 | 134 | 5.6% |
| FBPNN-PID | 9 | 14 | 50 | 3.5% |
| IPSO-FBPNN-PID | 11 | 13 | 22 | 0.2% |

The pH value was reduced from 6.9 to 5.3, with the performance parameters shown in Table 4. Compared to the first three algorithms, the controller in this paper reduced the amount of overshoot by 13.8, 5.4 and 3.3 percentage points. The IPSO-FBPNN-PID controller can, therefore, respond to the set pH value in a short time with minimal overshoot and can meet the control requirements.

## 7. Conclusions

In this study, a neural network PID controller with improved particle swarm optimization network parameters and fuzzy preprocessed input signals was proposed to achieve a better response to the problems of nonlinearity and hysteresis of the fertilizer system and, thus, improve the accuracy of pH regulation of the fertilizer solution.

Through the simulation and practical application of pH control in the aqueous fertilizer solution, the excellent dynamic performance of the controller in this paper was verified step by step by comparing the data of response time, regulation time, rise time and overshoot of the four controllers with the current common control methods (PID control, fuzzy control) and fuzzy preprocessed neural network PID control.

The experimental results showed that the BP neural network PID algorithm based on improved particle swarm and fuzzy preprocessing optimization had better dynamic performance. Compared to the other three algorithms, the IPSO-FBPNN-PID control algorithm was significantly better in terms of overshoot and steady-state time. Additionally, the steady-state performance was also better. This controller is able to reduce the effects of time lag and nonlinearity in the actual fertilizer application process, which can produce better satisfying results in the work of fertilizer pH-worthy regulation within the fertilizer application system. In the future, the optimization of other intelligent algorithms and the combination of advantages with each other will also be further considered to greatly adapt to more complex regulation processes.

**Author Contributions:** This study was conceptualized by R.Z. and L.Z. The software was designed by R.Z. and validated by Z.M., C.D. and C.F. R.Z. provided resources and R.Z. curated the data. The original draft of the manuscript was prepared by R.Z. and C.F. H.W. reviewed and edited the manuscript. C.D., Y.S. and H.B. assisted with project administration. H.W. and Z.M. managed funding acquisition. All authors have read and agreed to the published version of the manuscript.

**Funding:** This research was funded by the National Natural Science Foundation of China, grant number 52065055, and Provincial and Ministerial Projects No. 2021JS004.

**Institutional Review Board Statement:** Not applicable.

**Informed Consent Statement:** Not applicable.

**Data Availability Statement:** All relevant data presented in the article are stored according to institutional requirements and, as such, are not available on-line. However, all data used in this manuscript can be made available upon request to the authors.

**Acknowledgments:** We are grateful to Changxin Fu, Zihao Meng and Huan Wang for their data recording and checking work during the process. We are also grateful to Chanchan Du, Yongchao Shan and Haoran Bu for their help in project management. We also thank Shihezi University for providing the experimental conditions for us to successfully complete this experiment. Finally, we thank the instructor for his constructive comments on the earlier version of the manuscript.

**Conflicts of Interest:** The authors declare no conflict of interest.

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
