# Peer review of "Fuzzy Neural Network PID Strategy Based on PSO Optimization for pH Control of Water and Fertilizer Integration"

_applsci, doi:10.3390/app12157383_

Round 1

Reviewer 1 Report

The authors have developed a system to control the PH level of water and fertilizer mixture. They proposed a hybrid control of fertilizer and water conditioning systems, then builds a fuzzy pre-pro-cessing controller and a neural network proportional-integral differential controller. Additionally, they optimized the neural network parameters by means of an improved particle swarm algorithm.

My question as a farmer is that.

1-

Determining the PH value of fertilizer and water is a once activity. once we get to know about the good mixture of fertilizer and water we are repeating the same every time and the problem is solved forever unless fertilizer is changed or water is changed or soil condition is changed. changing any of these parameters is not so frequent (not changing every minute or even hours some time months).

in this case, why do we need AI, nural network, and fuzzy logic to control al these things?

2-

language is needed to improve some suggestions are given in the attached file. I recommend checking whole the article once again from the perspective of language. 

Reviewer 2 Report

Authors have discussed the Fuzzy-neural network-PID strategy based on PSO optimization for pH control of water and fertilizer integration

1.       It is strongly recommended to discuss the related work of the proposed solutions.

2.       There are some grammatical mistakes such as after “,” space is not provided for example: In the title of the figure1. Therefore, please proofread the paper one more time.

3.       Results analysis part must be improved with a detailed explanation for each diagram 

Round 2

Reviewer 1 Report

i recommend this paper for publication